# Comparative Analysis of Bone Ingrowth in 3D-Printed Titanium Lattice Structures with Different Patterns

**DOI:** 10.3390/ma16103861

**Published:** 2023-05-20

**Authors:** Ágnes Éva Kovács, Zoltán Csernátony, Loránd Csámer, Gábor Méhes, Dániel Szabó, Mihály Veres, Mihály Braun, Balázs Harangi, Norbert Serbán, Lei Zhang, György Falk, Hajnalka Soósné Horváth, Sándor Manó

**Affiliations:** 1Laboratory of Biomechanics, Department of Orthopaedic Surgery, Faculty of Medicine, University of Debrecen, H-4032 Debrecen, Hungary; csz@med.unideb.hu (Z.C.); csamer.lorand@med.unideb.hu (L.C.); szabodaniel@med.unideb.hu (D.S.); leizhang@med.unideb.hu (L.Z.); soosne.hajnalka@med.unideb.hu (H.S.H.); manos@med.unideb.hu (S.M.); 2Department of Pathology, Faculty of Medicine, University of Debrecen, H-4032 Debrecen, Hungary; gabor.mehes@med.unideb.hu; 3Isotoptech Private Limited Company, H-4026 Debrecen, Hungary; veresmihaly@isotoptech.hu (M.V.); mbraun@atomki.hu (M.B.); 4Department of Data Science and Visualization, Faculty of Informatics, University of Debrecen, H-4028 Debrecen, Hungary; harangi.balazs@inf.unideb.hu (B.H.); serban.norbert@inf.unideb.hu (N.S.); 5Varinex Private Limited Company, H-1141 Budapest, Hungary; falk@varinex.hu

**Keywords:** additive manufacturing, direct metal laser sintering, Ti6Al4V, lattice structure pattern, bone ingrowth, sheep, osseointegration

## Abstract

In this study, metal 3D printing technology was used to create lattice-shaped test specimens of orthopedic implants to determine the effect of different lattice shapes on bone ingrowth. Six different lattice shapes were used: gyroid, cube, cylinder, tetrahedron, double pyramid, and Voronoi. The lattice-structured implants were produced from Ti6Al4V alloy using direct metal laser sintering 3D printing technology with an EOS M290 printer. The implants were implanted into the femoral condyles of sheep, and the animals were euthanized 8 and 12 weeks after surgery. To determine the degree of bone ingrowth for different lattice-shaped implants, mechanical, histological, and image processing tests on ground samples and optical microscopic images were performed. In the mechanical test, the force required to compress the different lattice-shaped implants and the force required for a solid implant were compared, and significant differences were found in several instances. Statistically evaluating the results of our image processing algorithm, it was found that the digitally segmented areas clearly consisted of ingrown bone tissue; this finding is also supported by the results of classical histological processing. Our main goal was realized, so the bone ingrowth efficiencies of the six lattice shapes were ranked. It was found that the gyroid, double pyramid, and cube-shaped lattice implants had the highest degree of bone tissue growth per unit time. This ranking of the three lattice shapes remained the same at both 8 and 12 weeks after euthanasia. In accordance with the study, as a side project, a new image processing algorithm was developed that proved suitable for determining the degree of bone ingrowth in lattice implants from optical microscopic images. Along with the cube lattice shape, whose high bone ingrowth values have been previously reported in many studies, it was found that the gyroid and double pyramid lattice shapes produced similarly good results.

## 1. Introduction

The number of metal implants that can be implanted into the human body is growing. Due to increasingly high quality requirements, the performance of these implanted metals has greatly improved. Today, implants intended for human use need to meet demanding requirements with respect to both metallurgical and biological attributes. One of the most basic performance expectations concerns bone incorporation [1,2].

Many researchers agree that, among the metals currently available, titanium and its alloys are the most suitable for human use in many body parts [1,3,4,5]. Ti6Al4V alloy has many desirable implantability properties, including excellent biocompatibility, high levels of osseointegration and corrosion resistance, and a combination of mechanical modulus and strength that is similar to that of bone, all of which are obtained without cytotoxicity or other negative effects on the human body [6,7,8,9]. In addition, Ti6Al4V alloy is relatively easy to obtain, and there is a large body of scientifically validated knowledge regarding its human use, along with considerable experience of its alloying and industrial processing.

The characteristics of a favorable pore size are as follows: it should allow for the fastest and most extensive bone growth, and should enable both the highest maturity of the bone and the most extensive filling of the pores by the bone [10,11,12]. The size, shape, type, and location of the pores significantly affect the osteoconductive effect of the implant. Of these, the effect of pore size has been studied most intensively [12,13,14,15,16,17,18].

A minimum pore size of 100 µm is required for osteoblast ingrowth [13,16]. The purpose of another study was to evaluate the osseointegration of porous titanium implants coated with autologous osteoblasts. Titanium implants with 400, 500, and 600 pm diameter drill channels were coated with autologous osteoblasts obtained from cancellous bone chips. The implants were placed in the distal femora of rabbits. The 600 pm channel showed the best ingrowth of bone tissue [10]. Another study presents an experiment using bone marrow mesenchymal stem cells to study the proliferation and differentiation of cells on titanium scaffolds with different pore sizes [12]. In the study by Buj-Corral et al., a model was developed to define the pore size and porosity of porous structures. The model was applied to a disk shape. In order to compare the experimental results with the computational results, samples were printed using FDM technology. A random trabecular structure was printed [19]. Pore sizes of 300, 600, and 900 μm were produced by SLM. A diamond lattice was adapted as the basic structure.

Cylindrical porous titanium implants were implanted into the cancellous bone of the rabbit femur. Because of its adequate mechanical strength, high fixation ability, and rapid bone ingrowth, our results indicate that the pore structure of the 600 μm implant is a suitable porous structure for orthopedic implants manufactured by SLM [20].

Further studies have reported rapid and complete ingrowth into pores with a diameter of 600 µm, which has been confirmed by histological examination [17,18]. At this pore size, the formation of a well-mineralized intercellular matrix can be expected, as well as the provision of adequate nutrient and oxygen transport and the development of vascularization, which is necessary for cell growth [14]. Overall, numerous studies have concluded that ossification is most intensive at a pore diameter of 500–600 µm [10,17] (Table 1).

As can be seen from the above, the information on bone ingrowth in lattice structures is mostly related to pore size; we know less about different shapes of lattices tested under the same conditions with the same pore size. In this study, we aimed to analyze the effects of different lattice shapes on bone ingrowth using innovative methods to find structures to be suggested for biologically fixing titanium implants. We examined six different lattice test specimens in animal experiments using sheep, and we conducted mechanical, histological, and image processing analyses on the samples obtained. The samples were cut using the novel water jet method, and we applied a newly developed image processing method to calculate the bone ingrowth.

## 2. Materials and Methods

### 2.1. Design and Manufacture of Implants

We investigated the following shapes in our animal experiments: gyroid, cube, cylinder, tetrahedron, double pyramid, and Voronoi (Figure 1). Some of these shapes were chosen for comparison with those examined by several authors [6,8,21,22,23,24,25,26,27,28]. The degree of bone growth was determined using rivet-like cylindrical implants with a diameter of 6 mm and a rounded shape; these were used in the animal experiments. The head portion was 2 mm high and had a diameter of 10 mm. Due to manufacturing considerations, the center of the implants was solid, with a diameter of 2 mm. The pore size of the implants was consistently 600 µm. The lattice-structured implants were produced using Spaceclaim 2019 (Ansys, Canonsburg, PA, USA) and Element (nTopology Inc., New York, NY, USA) software (Figure 1).

The implants were manufactured using an EOS M290 (EOS GmbH, Krailling, Germany) 3D printer, employing direct metal laser sintering technology with Ti6Al4V (Grade 23) material. Implants were placed perpendicular to the platform, while their heads were in contact with the surface of the printing platform without any support structures. The main parameters of laser sintering are recorded in Table 2. After 3D printing was completed, rivets were cut off from the platform using a band saw, and then a 0.5 mm chamfer was applied to their sawn edges as a deburring method (Figure 1).

### 2.2. Animal Experiment and Surgical Technique

The animal experiment was carried out in the operating room of the Experimental Surgical Institute of the Semmelweis University in Herceghalom, with an ethics reference number of PE/EA/573-8/2019. For ethical reasons, we worked with the smallest possible number of experimental animals that still allowed for statistical evaluation.

Our study consisted of 12 Merino ewes aged 3–4 years and weighing 45–50 kg. We implanted three implants into the two femoral condyles of each sheep. One of the three implants was always a control, which was solid and without a trabecular structure. The other two were lattice-structured implants of different lattice types (Figure 2). The same lattice types were implanted on the other side.

Surgical implant placement procedures were performed under intratracheal narcosis. All practices commonly observed in surgery on humans were adhered to, including gowning and regular surgical scrubbing, and full attention was paid to sterility throughout.

We made an oblique incision on the inner surface of the femoral condyle, partly along the fibers of the vastus medialis and partly cutting through them to penetrate. We then exposed the medial cortical of the condyle. After exposure, we attached one leg of the custom-made targeting pin—which we had developed and 3D printed—to the medial femoral condyle, and attached the other pin to the accessible edge of the medial femoral condyle. Placing the targeting device on the exposed bone surface, we drilled through the full width of the femoral condyle using a 6 mm diameter drill. We then press-fitted the implants into the three prepared holes in an easily identifiable manner (Figure 3). Finally, we closed the wound with interrupted sutures. Each planned implantation was successful in every experimental animal.

By subjecting animals to two rounds of prolonged anesthesia, at 8 and 12 weeks following implantation, we were able to study the process of osseointegration. Femur condyle pieces containing the implants were fixed in formalin. A total of 72 implants were inserted into the animals, and their distribution is listed in Table 3. To assess the degree of bone growth, half of the removed femurs were examined histologically, while the other half underwent mechanical testing.

### 2.3. Mechanical Evaluation

We conducted push-out tests to determine the force required to displace the implants with different lattice structures from their positions, and to measure the resistance created by bone growth. The greater the force required for displacement, the greater the assumed degree of bone growth. As the cylindrical implants created a hole completely through the bone during surgery, it was possible to displace the implant from the hole in the opposite direction of insertion, similar to Pobloth’s study [24] (Figure 4).

The tests were performed using an Instron 8874 materials testing machine (Instron, Norwood, MA, USA) in our Biomechanical Materials Testing Laboratory. A 5 mm diameter stainless steel cylindrical rod was fixed in the upper grip, and a disc with a centrally drilled hole was used to resist the displacement of the implant under the bone. During the test, the displaced implant reached the hole in the disc. The testing speed was 1 mm/minute in all cases. Each test was performed until the implant was completely pushed out from the specimen.

### 2.4. Histological Processing

The implants were removed from the femur condyles with approximately 5 mm of their bony surroundings. Longitudinal sections through the axis of the implants needed for further processing were made using an I510-G2 waterjet cutting machine (TECHNI Waterjet, Campbellfield, Australia).

The sections were cleaned in isopropanol using a Sonorex RK100H (Bandelin, Berlin, Germany) ultrasonic cleaner. They were then ground and polished on an Ecomet II Grinder Polisher (Buehler, Lake Bluff, IL, USA) grinding wheel used for preparing geological samples. An example of a prepared sample is shown in Figure 5, where the bone tissue, scarring, ingrowth, and implant structure are clearly visible.

Optical microscopic images of the sections were taken using a Keyence VHX-6000 (Keyence International, Mechelen, Belgium) digital microscope. VH-Z20R/Z20T objectives were used to capture the images. The degree of bone growth was evaluated using a newly developed algorithm and by comparisons of manual drawings on the microscopic images.

To confirm and support the results of the image processing method, we also examined the histomorphology of the implants following removal, and carried out histological processing of tissue pieces both from the surface and from the inner cavitations of the implants. The samples were processed and embedded in paraffin for analysis after decalcification in EDTA according to routine histological protocol. The tissue composition was determined using standard 4 µm thick hematoxylin–eosin (H&E)-stained sections. Morphological analysis was carried out using light microscopy (Leica DM3000 LED microscope, Leica Microsystems GmbH, Wetzlar, Germany), and images were obtained using an integrated digital camera (Leica Flexacam C1, Leica Microsystems GmbH, Wetzlar, Germany) and LAS X v3.0.13 imaging software (Leica Microsystems GmbH, Wetzlar, Germany). 

### 2.5. Image Processing Algorithm

A computer-based image processing method was developed to measure the quantity of newly grown bone tissue that appeared on the microscopic images of the implants. This automated process efficiently measures the efficacy of different types of grid structures applied to the implants. The method consists of two basic steps. The first step involves segmenting the implants by extracting them from the background. To accomplish this, the input color image (shown in Figure 6a) is converted to grayscale, and a mean thresholding algorithm [25] is used to remove most of the background (Figure 6b). A sliding-window algorithm is then employed to eliminate the remaining background pixels. To achieve this, a Sobel edge filter [26] is applied to the resulting image (Figure 6c), and the sliding-window algorithm classifies each subregion as either foreground or background. If the sliding window contains a sufficient number of hard-edge pixels, it is considered to be foreground; otherwise, it is considered background. This step produces a binary image, as shown in Figure 6d, which can be used to extract the implant region. As a refinement, a morphological closing operator is applied to smooth the edges and correct segmentation errors. The result of this segmentation algorithm is shown in Figure 6e.

In the second part of the image processing, we needed to identify three different textures in the implant region: the pure surface of the implant, the osseointegrated area, and the holes of the grid structure that were not osseointegrated. To achieve this, as a first step, we used an active contour model to determine the most likely boundary of these holes. The active contour method allows the contour to vary iteratively to minimize the well-known Chan–Vese energy function [27]. The advantage of this energy function (1) is that it can segment these holes with smoother boundaries. The evaluation of the curve depends only on the difference in pixel intensities (*I_x,y_*) and the average intensities inside (*c*_1_) and outside (*c*_2_) the curI (*C*), as formulated in (2).
(1)$$E_{Chan\_Vese}=F_{1}\left(C\right)+F_{2}\left(C\right),$$
(2)$$F_{1}\left(C\right)=\int_{inside\left(C\right)}^{}{\left|I_{x,y}-c_{1}\right|}^{2}dxdy,$$
$$F_{2}\left(C\right)=\int_{outside\left(C\right)}^{}{\left|I_{x,y}-c_{2}\right|}^{2}dxdy.$$

After all of the existing holes are recognized, the remaining regions can be classified as implant or bone pixels by analyzing the roughness of their texture. The roughness is derived from the Roberts edge filter [28]. Using calculated Roberts gradients, a pixel and its local neighborhood can be considered a bone (or implant) region if the average gradients are higher (or lower) than a previously set threshold value. Figure 7 illustrates the steps of this second segmentation process.

To predict the effectiveness of different types of grid structures, and to calculate how well they aid the osseointegration process, we measured the bone ingrowth ratio (*BI*) between the segmented areas of the osseointegrated region (*A_Oss_*) and the original number of hole regions, using the following formula:$$BI=\frac{A_{Oss}}{A_{Oss}+A_{Hole}},$$
where *A_Hole_* refers to the detected hole regions.

By calculating the *BI* ratio, the algorithm approximates the level of osseointegration for different grid structures, as can be seen in the examples presented in Table 4.

### 2.6. Manual Image Processing

Alongside the analysis of the image processing algorithm, we also manually determined the level of bone ingrowth in the images using Image J software (National Institutes of Health, Bethesda, MD, USA) through manual outlining. The images were processed by two independent individuals, and the quantified results obtained were compared to the data obtained from the image processing algorithm. Figure 8 shows an example of the results of manual image processing and the resulting BI value.

### 2.7. Statistical Analysis

In this study, we tested two hypotheses:There is no significant difference in the push-out force between the solid grid and the other grid types.There is no significant difference in ingrowth between the examined grid types at week 8 and week 12, and the ingrowth over time within each grid type does not change significantly.

We assessed the normality of the population using the Shapiro–Wilk test. We analyzed statistical differences between different grids and the solid grid using one-way ANOVA supplemented with Dunnett’s post hoc test. We used a linear regression model to compare the algorithm method and the manual method. We used two-way ANOVA complemented by Sidak’s multiple comparison test to assess the statistical differences between the investigated grids at different time points. We considered *p* < 0.05 to be statistically significant.

We present the data as the mean ± 95% confidence intervals. We performed statistical calculations using GraphPad Prism 7 (GraphPad Software Inc., San Diego, CA, USA).

## 3. Results and Discussion

Thanks to the work of the Swiss Arbeitsgemeinschaft für Osteosynthesefragen (AO) working group, sheep have become the gold standard for bone surgery experiments worldwide. AO experiments have shown that the results obtained using sheep can be applied to human bone surgery in many respects [29]. Because the spongy bone of the distal femur condyle of sheep most resembles that of humans at a young age, i.e., 3–4 years old, we used sheep of this age in our animal experiment.

### 3.1. Mechanical Evaluation

We compared the push-out force of implants with different grid types to that of the solid control implant (Figure 9). The push-out force of the gyroid (*p* = 0.019) and cube (*p* = 0.003) grid types was significantly higher compared to that of the solid control implant. In the case of the cylindrical grid, a borderline significance was obtained compared with the control (*p* = 0.053). The push-out force was also higher in the case of the double pyramid grid, although the difference was not significant (*p* = 0.215). There were no significant differences in the push-out force between the tetrahedron (*p* = 0.884) or Voronoi (*p* = 0.997) grids and the solid grid. Overall, because there were several significant differences compared to the control, we rejected our first hypothesis (see Section 2.7). At the same time, we can see that the presence of bone tissue alone does not guarantee necessarily better mechanical properties, because the mechanical strength is determined by the quality, not the quantity, of the bone.

### 3.2. Comparison of the Results of the Image Processing Algorithm and Manual Drawing

A comparison between the algorithm and manual methods was carried out using a linear regression model. We compared the data generated by the algorithm to the manually input data. The two different measurements showed a very strong correlation (R^2^ = 0.924; *p* < 0.001). Examples of manually drawn and algorithmically drawn cube grid tests are presented in Figure 10.

During histological examination, we invariably observed signs of normal bone formation, with slightly irregular but well-developed pseudo-lamellar bone tissue detectable in all cases. The locations of the titanium mesh appear as empty holes in the sections, surrounded mostly by newly formed bone, with little adipose or connective tissue, and minimal bone marrow filling the cavities, without any signs of irritation or inflammation (Figure 11).

All of these observations support the results obtained using the image processing algorithm, indicating that the digitally segmented and measured areas on the water-cut surface can be considered relatively homogeneous and properly formed mature bone based on tissue composition.

### 3.3. Bone Ingrowth Efficiency in Terms of Lattice Type and Time Elapsed after Implantation

We ranked the bone ingrowth efficiency of the six types of lattice implants according to the above results.

In the case of the tetrahedron, ingrowth was significantly lower compared to all other grid types both at week 8 (tetrahedron vs. Voronoi, *p* = 0.005; tetrahedron vs. cylindrical, gyroid, double pyramid, cubic, *p* < 0.001) and at week 12 (tetrahedron vs. Voronoi, cylindrical, gyroid, double pyramid, cubic, *p* < 0.001). Although the Voronoi grid showed significantly greater ingrowth than the tetrahedron grid, compared to all other grid types, its performance was significantly lower in terms of ingrowth both at week 8 (*p* < 0.001 in Voronoi vs. cylindrical, gyroid, double pyramid, cubic) and at week 12 (*p* < 0.001 in Voronoi vs. cylindrical, gyroid, double pyramid, cubic). There was no significant difference between the cylindrical and the gyroid grids either at week 8 (*p* = 0.742) or at week 12 (*p* = 0.995). While there was no significant difference between the cylindrical and the double pyramid grids at week 8 (*p* = 0.113), the double pyramid grid showed significantly greater growth at week 12 (*p* = 0.044). At both week 8 (*p* = 0.037) and week 12 (*p* = 0.003), the cubic grid showed greater ingrowth compared to the cylindrical grid. Furthermore, we did not find a significant difference between the gyroid and the double pyramid grids at any time (week 8, *p* = 0.857; week 12, *p* = 0.138). There was no significant difference between the gyroid and the cubic grids at week 8 (*p* = 0.423); however, in the case of week 12, we saw that the ingrowth was significantly greater in the case of the cubic grid (*p* = 0.01). Finally, there was no significant difference between the double pyramid and the cubic grids at any time point (week 8, *p* = 0.889; week 12, *p* = 0.443).

In terms of bone ingrowth, the most effective lattice types were ranked as follows: gyroid, double pyramid, and cubic. The order of these three lattice types was the same for both weeks, i.e., the ranking order remained the same at both week 8 and week 12 after euthanization. For this reason, we also rejected our second hypothesis (see Section 2.7).

During the time comparison, only in the case of the Voronoi grid was there a significant difference between week 8 and week 12 (*p* < 0.001); for the other grid types, there were no significant differences (tetrahedron, *p* = 0.999; cylindrical, *p* = 0.648; gyroid, *p* = 0.999; double pyramid, *p* = 0.088; cubic, *p* = 0.078) (Figure 12).

The requirements for bone joint implants may vary depending on the body region and the purpose of the application. For long-term endoprostheses, fusion of the bone implant by bone ingrowth is desirable. In these cases, there is a role for bone-growth-promoting resurfacing. In recent studies [7,9,20,21,30,31,32], the use of various lattice structures, especially for custom-made implants that can be produced by 3D printing, has become increasingly prevalent. The determination of the ideal lattice size for bone growth can be rather clearly defined based on relevant studies in the literature [12,15,16,17,18,20,21]. However, the effect of the lattice shape on bone growth still raises many unanswered questions. This was the motivation for the current study.

We have taken a different approach to the experiments found in the literature. Six different types of lattice shapes were tested at a time, with a 12-week follow-up, and implants were placed in the unloaded area of the femur.

Our results are broadly in line with those reported in similar studies in the literature, but provide several new insights. Based on our experiments and results, we can state that there are major differences in bone ingrowth between the lattice shapes. However, the explanation is not unequivocal. The cell-scale biological process of bone growing in an in vivo metal lattice environment cannot be detected easily, but there are some factors that cause more significant ingrowth in the case of round-shaped and quasi-round-shaped holes, such as gyroid, double pyramid, and cube. This result similar to the conclusion of Deng, who found that the quasi-round DIA lattice shape demonstrated the best structural bone growth [33].

In terms of 3D-printer-fabricated lattice structures, there are mainly morphological, mechanical, finite element studies, where the authors investigated the strength of different shapes [2,7,34,35,36,37]. Several authors have studied the effects of pore size on bone ingrowth [11,12,13,14,15,16,17,18,19,20,21,22,23]. In this respect, we chose a size of 600 µm based on their work, which proved to be a correct selection in the experiments (Table 1). There are fewer studies that have already looked at bone ingrowth in vivo. Although the follow-up times differ (4–8 weeks vs. 8–12 weeks), Arabnejad’s work also shows that there are substantial differences in the bone ingrowth of each grid shape, while the ingrowth values are similar to the current research [38]. Chen at al. experimented with rats, where the specimen size is so small compared to human implants and pore sizes that it is difficult to make a correct assessment of bone ingrowth [39]. In Van Bael’s study, several lattice shapes were used; the differences between the ingrowth capabilities had been discovered similarly in this work, but direct comparison to our in vivo study is too difficult [21]. Biemond et al. tested only two different grids, but similar to our results, the cubic grid showed positive results after 6 weeks [40].

## 4. Conclusions

Based on the results of our experiments and investigations, we draw the following conclusions:The shape of the lattice structure of 3D-printed titanium implants has a great influence on bone ingrowth.The results of statistical and classical histological processing confirm that our newly developed image processing algorithms are suitable for the accurate and precise determination of the extent of bone growth.The lattice shapes containing round-shaped and quasi-round-shaped holes, i.e., the gyroid, double pyramid, and cube, facilitate more bone ingrowth. Additionally, considerably high bone ingrowth rates (60–80%) can be detected in the case of these most successful lattice types in sheep femoral condyles after 12 weeks.

In summary, our experiments were used to determine the indentation values and biological fixation strengths that could be achieved with each lattice shape in an unloaded environment without press-fit fixation. Our results hopefully can be used for the design of biologically fixed porous titanium implants.

## Figures and Tables

**Figure 1 materials-16-03861-f001:**
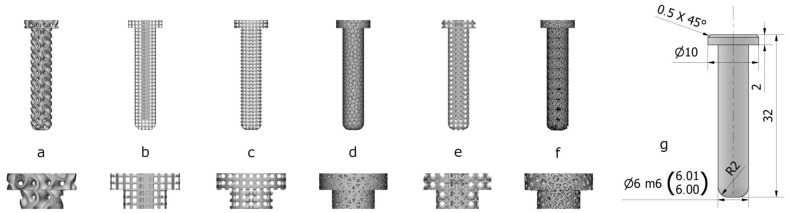
The specimen dimensions and lattice types used in the experiments: (**a**) gyroid; (**b**) cube; (**c**) cylinder; (**d**) tetrahedron; (**e**) double pyramid; (**f**) Voronoi; (**g**) solid.

**Figure 2 materials-16-03861-f002:**
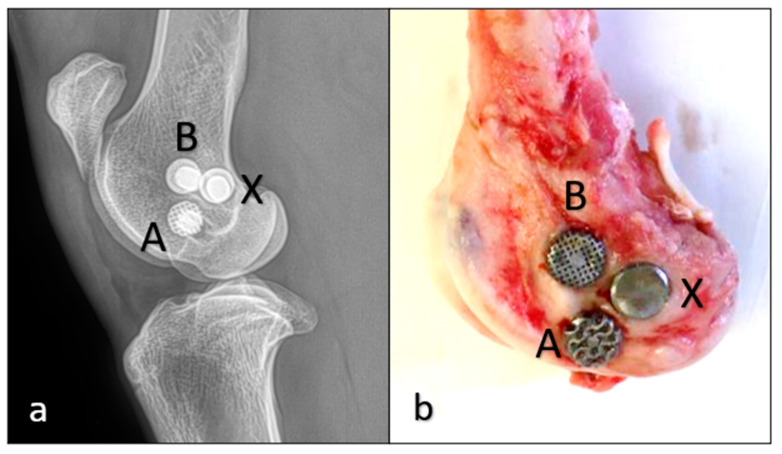
Implant placement in a femoral condyle: (**a**) an X-ray image taken after surgery; (**b**) a picture of a femoral piece with the implants 12 weeks after the surgery. “A”, gyroid lattice; “B”, cube lattice; “X”, solid, control.

**Figure 3 materials-16-03861-f003:**
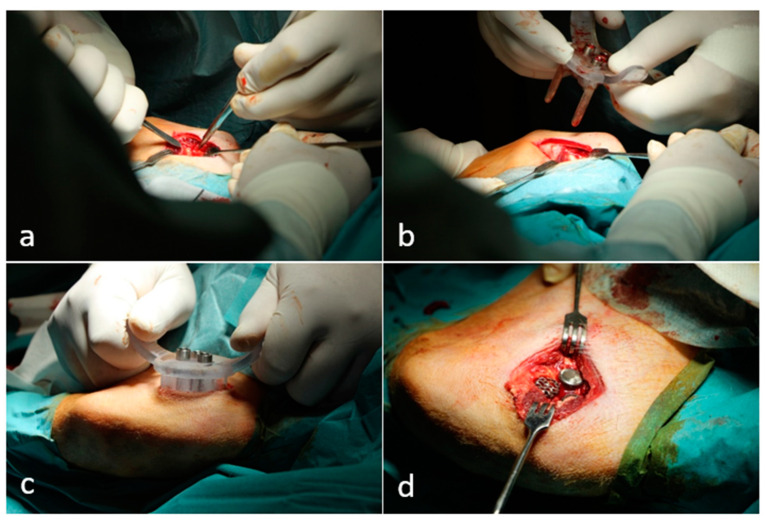
Main steps of the surgical procedure: (**a**) preparation of implant locations within the surgical area; (**b**) insertion of the targeting device; (**c**) preparation of holes using the surgical targeting device; (**d**) insertion of the implants.

**Figure 4 materials-16-03861-f004:**
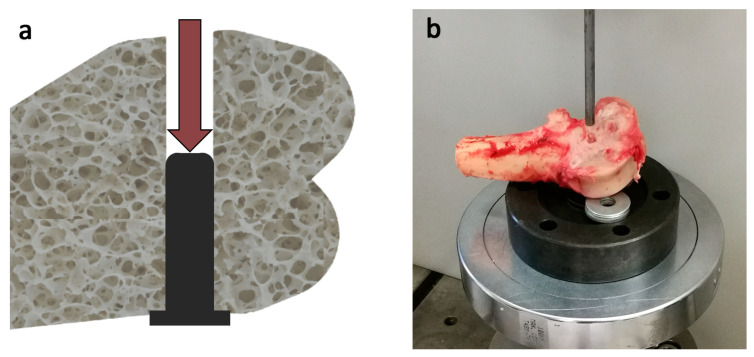
(**a**) Schematic diagram of the push-out test; (**b**) execution of the push-out test.

**Figure 5 materials-16-03861-f005:**
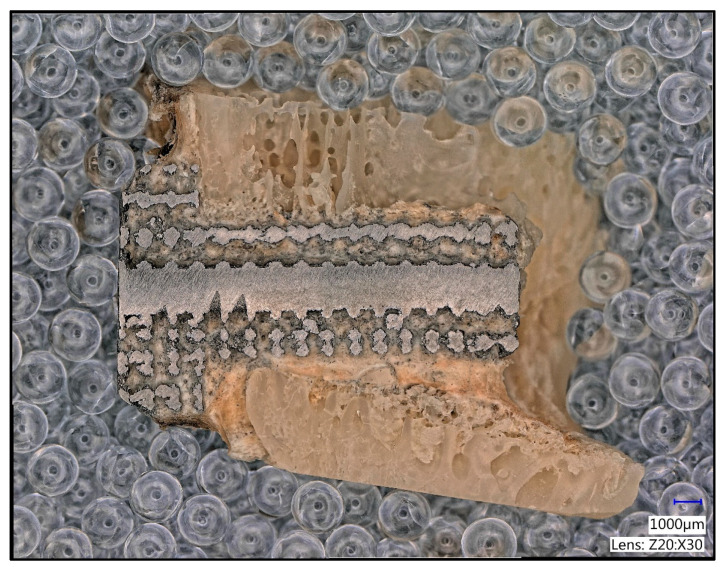
Longitudinal section of a 12-week bony femur condyle sample after cutting.

**Figure 6 materials-16-03861-f006:**
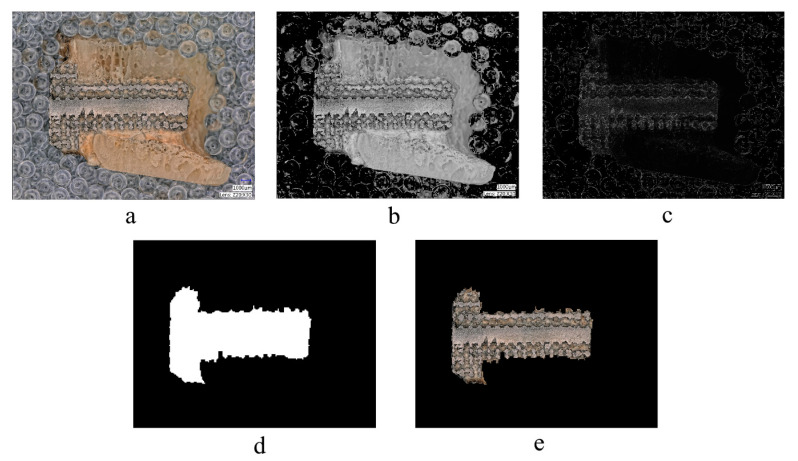
Steps of the image segmentation: (**a**) input image; (**b**) mean thresholded grayscale image; (**c**) edge map image; (**d**) binary segmentation mask; (**e**) segmented implant.

**Figure 7 materials-16-03861-f007:**
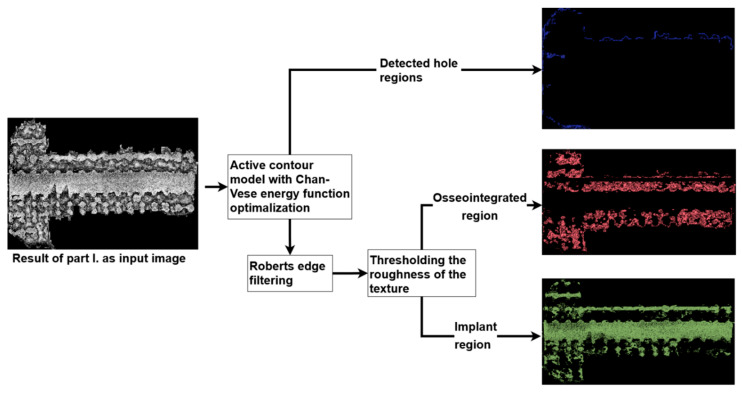
The steps of the second part of the image processing.

**Figure 8 materials-16-03861-f008:**
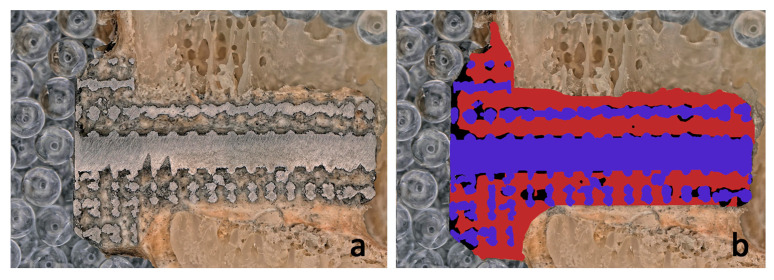
Analysis of the level of bone growth on (**a**) an optical micrograph image with a cube-grid-structured implant (**b**) through manual outlining. The BI ratio obtained using this method is 89.26%.

**Figure 9 materials-16-03861-f009:**
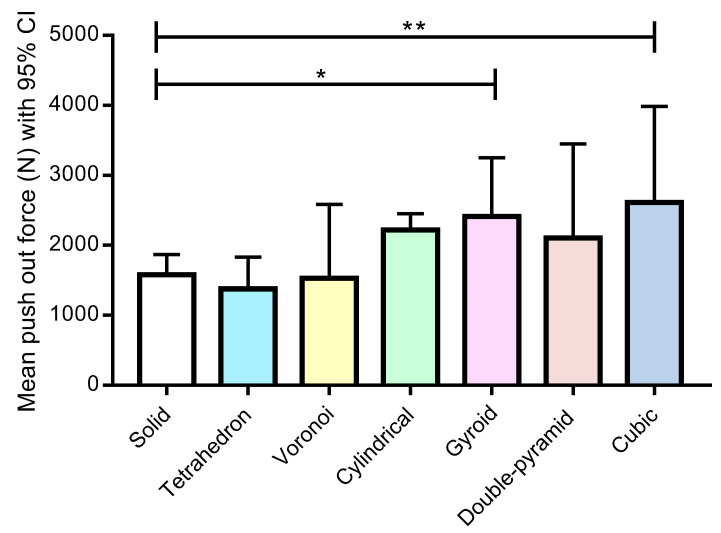
The push-out force (N—Newton) of different grids compared to that of the solid grid (*n* = 30). Data are presented as the mean ± 95% confidence intervals (95% CI). * and ** indicate statistically significant differences at *p* < 0.05 and *p* < 0.01, respectively.

**Figure 10 materials-16-03861-f010:**
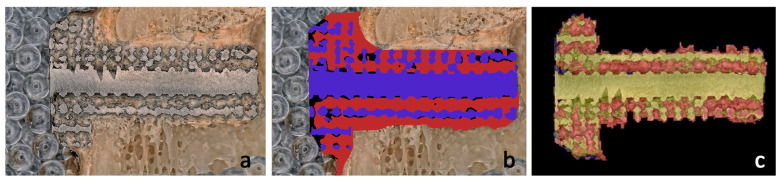
Analysis of bone ingrowth shown on an optical microscopic image of a cube grid test specimen: (**a**) image shows the original input picture; (**b**) image shows the result obtained by manual drawing; (**c**) image shows the result obtained by the image processing algorithm.

**Figure 11 materials-16-03861-f011:**
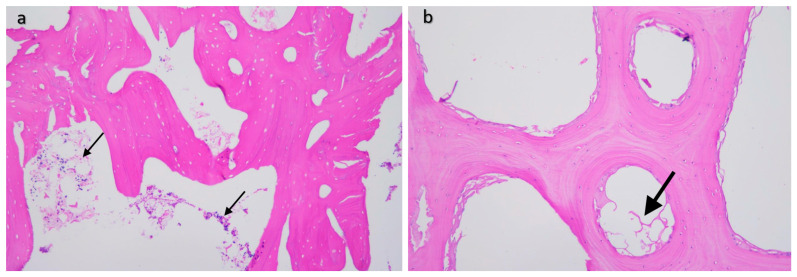
(**a**) Microscopic image of a 12-week gyroid-lattice-type implant processed by classical histological examination. Trabecular bone appears as eosinophilic (pink/red) material with somewhat irregular lamellar structuration, while some loose parenchymal cellular content consistent with fatty bone marrow fills the intertrabecular spaces (thin arrows). (**b**) Fully reconstructed bone presents with well-formed oriented lamellar architecture with mature adipose tissue in the intertrabecular spaces (fat arrow). (H&E staining, 200× magnification).

**Figure 12 materials-16-03861-f012:**
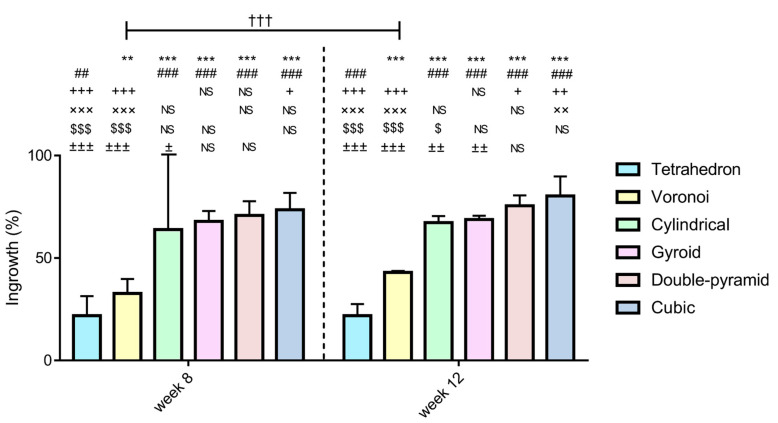
Statistical differences between different grid types at different time points. Data are shown as the mean ± 95% confidence intervals (95% CI). Comparisons of different grids at week 8 and week 12 time points are denoted by +/$/±; **/##/++/xx/±± and ***/###/+++/xxx/$$$/±±± indicating statistically significant difference at *p* < 0.05, *p* < 0.01 and *p* < 0.001, respectively (**, *** vs. tethradeon; ##, ### vs. Voronoi; +, ++, +++ vs. cylindrical; xx, xxx vs. gyroid; $, $$$ vs. double-pyramid; ±, ±±, ±±± vs. cubic). “NS” indicates no significant difference. A statistically significant difference at *p* < 0.001 between week 8 and week 12 is denoted by †††.

**Table 1 materials-16-03861-t001:** Data of 3D-printed porous scaffold studies and the current research. (n.a. = not applicable).

Fabrication Technique	Surgical Area/Experimental Type	Scaffold Design	Species	Pore Size (µm)	Author and Reference
n.a.	distal femur	titanium implants coated with autologous osteoblasts	rabbit	400–600	Frosch et al. [10]
n.a.	hip, intramedullary region	trabecular	canine, rabbit	>100	Marin et al. [13]
n.a.	trabeculae	no data	no data	200–300	Nouri et al. [14]
vacuum diffusion bonding of titanium meshes	plate (in vitro assay)	random	rat (stem cells)	300–400	Chang et al. [12]
FDM	finite element analysis (in vitro)	trabecular	n.a.	250–500	Buj-Corral et. al. [19]
SLM	dorsal muscles	channel implant (with four square channels)	canine	≥500	Fukuda et al. [17]
SLM	in vitro and in vivo (femur)	versatile porous scaffolds	rabbit	600	Ran et al. [18]
SLM	metaphysis of the tibia	diamond crystal lattice	rabbit	600	Taniguchi et al. [20]
SLM	plate (in vitro assay), mechanical compression testing, computational fluid dynamical analysis	triangular, hexagonal, rectangular	human (stem cells)	500	Van Bael et al. [21]
SLM	medial femur	octadense, gyroid, dode,	rabbit	1000	Lim et al. [22]
LPBF	cyclic potentiodynamic polarization, post corrosion analysis	gyroid	n.a.	no data, only porosity	Sharp et al. [23]
EMB	femur condylus	gyroid, cubic, double- pyramid, tetrahedron, Voronoi, cylindrical	sheep	600	current study

**Table 2 materials-16-03861-t002:** Main parameters of metal 3D printing.

Layer height	30 µm
Laser power	280 W
Laser speed	1200 mm/s
Hatch distance	0.14 mm
Energy input	55.56 J/mm^3^
Exposure pattern	No pattern
Hatch sorting	Time optimized
Hatch features	SkyWriting
Hatch rotation angle	67°
Hatch offset	0.015 mm
Exposure mode	Single exposure

**Table 3 materials-16-03861-t003:** Types of implanted implants and euthanasia data.

Sheep	Grid Shape of Implanted Implants	Time of Euthanasia
1.	“A” gyroid, “B” cube, “X” solid	8 weeks
2.	“A” gyroid, “B” cube, “X” solid	8 weeks
3.	“A” gyroid, “B” cube, “X” solid	12 weeks
4.	“A” gyroid, “B” cube, “X” solid	12 weeks
5.	“C” cylinder, “D” tetrahedron, “X” solid	8 weeks
6.	“C” cylinder, “D” tetrahedron, “X” solid	8 weeks
7.	“C” cylinder, “D” tetrahedron, “X” solid	12 weeks
8.	“C” cylinder, “D” tetrahedron, “X” solid	12 weeks
9.	“E” double pyramid, “F” Voronoi, “X” solid	8 weeks
10.	“E” double pyramid, “F” Voronoi, “X” solid	8 weeks
11.	“E” double pyramid, “F” Voronoi, “X” solid	12 weeks
12.	“E” double pyramid, “F” Voronoi, “X” solid	12 weeks

**Table 4 materials-16-03861-t004:** BI Ratio calculated for different images (12-week samples).

Type of Lattice	Cube	Gyroid	Double Pyramid
Input image	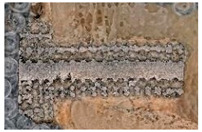	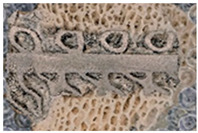	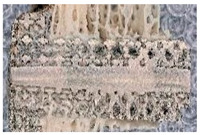
Result image	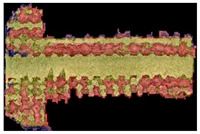	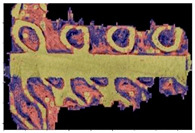	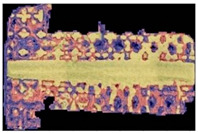
BI ratio	90.67%	66.77%	50.1%

## Data Availability

The data presented in this study are available upon request from the corresponding author.

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
