# Peer review of "Comparative Analysis of Bone Ingrowth in 3D-Printed Titanium Lattice Structures with Different Patterns"

_materials, 2023, doi:10.3390/ma16103861_

Round 1

Reviewer 1 Report

Major Comment:

11)      It looks like, there is a mismatch between the outcomes of Mechanical analysis (Push out force) and bone Ingrowth Efficiency. It is supposed to be collinear, meaning a higher push-out force indicates higher Bone ingrowth. But this does not hold true in some places between sections 3.1 and 3.3. Authors have to bring clarity on the lack of compatibility in the outcome.

22)      Comparison of outcome with other published articles is missing, authors have to add some explanation in the result and Discussion. Authors can select 2-3 most suitable references that match their work.

Reference 1) Bone Conduction Capacity of Highly Porous 3D-Printed Titanium Scaffolds Based on Different Pore Designs -2021

33)      Provide an explanation on why Gyroid and Cube design supports higher bone ingrowth compared to the tetrahedron and Voronoi etc. Authors can add an explanation in a suitable place before the conclusion after looking into their mechanical analysis, image processing, and other tests outcomes.   

Minor Comments:

44)      Authors can think of changing the title, as current title does not properly highlight the objective. Example: Comparative Analysis of Bone Ingrowth in 3D-Printed Titanium Lattice Structures of Different Patterns.

55)      In Page 9, section 3.1, what is the number of experiments conducted to get the deviation in bar diagram of figure 10, Mention ‘n’ value in the figure 10 title.

66)      Table 1, shows that tests are conducted post 8 and 12 weeks of implantation, it is not clear in any of the figures. As authors haven’t mentioned week details Except Figure 13.

77)      Page 2, line 79, It is written that “In addition to the lattice shapes most commonly reported in the literature [6,8,21–28], we investigated the following shapes in our animal experiments”, First part of the statement is not correct, it is meaning the author is investigating the lattice shapes which are already published in the literature.

88)      Figure -1, show the dimension on one of the schematic,

99)      It is better to add the figure of solid implant (control) in to the list of designs in the figure -1, you can show dimensions over it.

110)   Page 10, subsection 3.2, avoid writing single sentence paragraphs, author can merge it with above paragraph.

111)   Figure 10, reduce the font size of axis names and Axis title.

112)   Page 10, line 269 onwards till 277, Authors have written methodology-related content in result section can move this from here to Materials and methods, if not required.

113)   Page 11, line 103, replace the word ‘worse’ with a suitable one.

114)   Conclusions: avoid citing references

Reviewer 2 Report

The paper requires general language editing.

Reviewer 3 Report

The paper entitled “Bone Ingrowth into 3D-Printed Titanium Lattice Structures with Different Patterns" presents the use of metal 3D printing technology to create lattice-shaped test specimens of orthopedic implants to determine the effect of different lattice shapes on bone ingrowth. From my point of view, the topic is of great interest. But in general, need some reviews to be done:

·        The abstract provides a good overview of the research and its findings. But please reconsider using impersonal form of verbs. It can help to make your writing more objective and formal. (avoid We…/our…)

·        The introduction is poor and too many references are used references are used. For example [5,11,13,15,18,19], describe better the differences between the studies (perhaps the help of a table could help).

·        The end of the introduction focuses on what is shown as work rather than on novelty and motivation, which could be made clearer with a rewrite.

·        Could you add a scale in Figure 1.

·        I see some of the figures in a table, which is not the most appropriate.

·        I would like to know more about the manufacture of the rivets. I see little description of the additive manufacturing part.

·        Only some parameters of the AM process but tolerances or dimensions of the final part…

·        Conclusions should be based on some of the data shown in the results.

·        Please also add some information about future directions.

Reviewer 4 Report

Introduction, This investigation is a paper that presents information for researchers in the field of osseointegration of titanium implants. The design and microscopic characteristics of titanium implants can improve the bone. The size, shape, type, and location of the pores significantly affect the osteoconductive effect of the implant. The research on bone ingrowth in lattice structures is mostly related to pore size; but the influence of the design of shapes of lattices tested is less known.

The aim of the study must explain with more clarity.

Materials and methods.

The authors are defined each step fo research, and it defined the specific characteristics of  each subsection: Design and Manufacture of Implants, Animal Experiment and Surgical Technique, Mechanical Evaluation, Histological Processing, Image Processing Algorithm, and Statistical Analysis.

In the subsection 2.2. Animal Experiment and Surgical Technique, the first paragraphe (lines 96-102) must be changed to the section Discussion

The surgical procedure is an original protocol or it is based in an experimental previous study? In this case, the authors must include the reference.

The subsection 2.3. Mechanical evaluation must include references of the used methodology

The histological processing is an original protocol or it is based in an experimental previous study? In this case, the authors must include the reference.

Results and Discussion.

The first subsection must be the main objective of the experimental study, Bone ingrowth efficiency in terms of lattice type and time elapsed after implantation; and after Mechanical evaluation and Image processing algorithm and manual drawing.

Each subsection reported the findings of the experimental study, but there are not any discussion. The authors must analised the relevance of the results and must to compare with other experimental and updated studies.

The authors must improve the quality of figures.

Conclusions. This section must report the main and relevant aspects of the experimental study in a paragraphe.

The first paragraphe of this section is incorrect according to the results of the paper. Also it not necessary the inclusion of references. In fact, it must be eliminated.

The second paragraphe with 3 conclusions is a repetition of results.

Conclusively, the study is not ready for publication.

Round 2

Reviewer 3 Report

Some minor errors which I understand will be fixed with a final revision:

- Twice headed References.

- The numbering of the conclusions is wrongly introduced.

Reviewer 4 Report

The review is correct.